# Sustainable Engineered Design and Scalable Manufacturing of Upcycled Graphene Reinforced Polylactic Acid/Polyurethane Blend Composites Having Shape Memory Behavior

**DOI:** 10.3390/polym15051085

**Published:** 2023-02-21

**Authors:** Busra Cetiner, Gulayse Sahin Dundar, Yusuf Yusufoglu, Burcu Saner Okan

**Affiliations:** 1Integrated Manufacturing Technologies Research and Application Center & Composite Technologies Center of Excellence, Manufacturing Technologies, Sabanci University, Teknopark Istanbul, Istanbul 34906, Turkey; 2Faculty of Engineering and Natural Sciences, Materials Science and Nanoengineering, Sabanci University, Istanbul 34956, Turkey; 3Adel Kalemcilik Ticaret ve Sanayi A.S., Kocaeli 41480, Turkey

**Keywords:** shape memory polymer, blended polymer composites, upcycled graphene

## Abstract

Material design in shape memory polymers (SMPs) carries significant importance in attaining high performance and adjusting the interface between additive and host polymer matrix to increase the degree of recovery. Herein, the main challenge is to enhance the interfacial interactions to provide reversibility during deformation. The present work describes a newly designed composite structure by manufacturing a high-degree biobased and thermally induced shape memory polylactic acid (PLA)/thermoplastic polyurethane (TPU) blend incorporated with graphene nanoplatelets obtained from waste tires. In this design, blending with TPU enhances flexibility, and adding GNP provides functionality in terms of mechanical and thermal properties by enhancing circularity and sustainability approaches. The present work provides a scalable compounding approach for industrial applications of GNP at high shear rates during the melt mixing of single/blend polymer matrices. By evaluating the mechanical performance of the PLA and TPU blend composite composition at a 9:1 weight percentage, the optimum GNP amount was defined as 0.5 wt%. The flexural strength of the developed composite structure was enhanced by 24% and the thermal conductivity by 15%. In addition, a 99.8% shape fixity ratio and a 99.58% recovery ratio were attained within 4 min, resulting in the spectacular enhancement of GNP attainment. This study provides an opportunity to understand the acting mechanism of upcycled GNP in improving composite formulations and to develop a new perspective on the sustainability of PLA/TPU blend composites with an increased biobased degree and shape memory behavior.

## 1. Introduction

Shape memory polymers (SMPs) are an emerging class of smart polymers with remarkable potential for use in several industries since their intrinsic alignment collaboratively responds to various external and environmental stimuli [1]. To date, SMPs have found applications in numerous industries, for instance, electronics (as smart tools, flexible devices, and information carriers), biomedical, textile, robotics, and aerospace [2]. SMPs can change their original shape into a temporary shape when stimuli are applied, including light [3], pH [4], electrical fields [5], temperature [6], moisture [7], solvent [8], and magnetic fields [9]. Shape change in these polymers can occur due to both hard and soft segments in the structure. The hard/crystalline segments are responsible for preserving and memorizing the permanent shape and act as net points, while the soft/reversible segments act as switching segments and result in the recovery of the shape from the permanent to the temporary stage. SMPs have various properties, including high flexibility with fatigue performance, ease of processing, and lightness due to their intrinsic characteristic structures [10]. Among the several types of SMPs, thermal stimuli activate thermally induced SMPs, and the activation of shape change is initiated by a programming temperature (Tperm) for preserving the permanent shape and a transition temperature (Ttrans) for temporary shape recovery [1,11,12,13]. Herein, Ttrans can be either glass transition (T*g*) or melting (Tm) temperatures to program the shape recovery of activation [14].

Thermoplastic polyurethane (TPU) is one of the widely utilized polymers in thermally induced SMPs. TPU is an (AB) n-type block copolymer elastomer consisting of hard segments and soft segments, which are polyesters and polyols, respectively [15]. TPU has received great interest in industrial applications due to its highly diverse mechanical, resistance, and shape memory properties, including outstanding flexibility, superior toughness, high durability, and excellent biocompatibility [14,16,17,18,19,20,21,22,23,24,25]. On the other hand, there are some drawbacks of TPU, such as low T*g* and high decomposition temperatures [15]. In order to enhance its thermal and mechanical properties, there have been numerous attempts made at the blending process of TPU with other polymers such as polylactic acid [26], polycaprolactone (PCL) [27], lignin [28], and polymethyl methacrylate (PMMA) [29]. The development of biobased polymer blends with TPU has received great attention to enhance sustainability and provide more circular approaches in the field of SMPs. Currently, PLA is a good candidate for developing blend formulations with TPU since it is a biobased and biodegradable polymer derived from renewable sources, demonstrating elevated mechanical, thermal, and biofriendly properties, as well as benefiting from higher processing temperatures and shape memory characteristics [30,31,32]. In the literature, there are some studies focusing on the benefits and compatibility of TPU and PLA compositions. For instance, Jing et al. investigated the mechanical and shape memory properties of a PLA-TPU blend polymer having 80 wt% PLA and increased the elongation at break by 400%, shape recovery by 94%, and shape fixing ratios in tensile test mode by 82% [26]. In another work, Lai et al. demonstrated a 93.5% shape recovery ratio by blending PLA with TPU (50/50 by weight), resulting in the enhancement of its shape memory characteristics [30]. Furthermore, Ahmed et al. studied the shape memory characteristics of TPU-PLA foams by showing that an 80 wt% TPU blend retained its shape 3.4 times better than neat TPU [33]. Consequently, the blending process for TPU shape memory applications can enhance functionality and mechanical properties such as stiffness and strength and stress recovery.

Nanointegration in blending studies carries significant importance to enhance interfacial interactions within polymer phases to gain multifunctionality in composite structures [34]. However, there are some challenges in the incorporation of nanofillers with polymer chains and the occurrence of weak interfacial networks between nanofillers and the polymer matrix [35,36,37,38]. In recent years, graphene and its derivatives have been widely studied as main and co-reinforcements in various thermoplastic polymer matrices via different routes, including melt compounding, solvent-based mixing, and in situ polymerization [39]. Melt compounding is the most preferred and powerful technique in the industry owing to its quick and lower cost since it does not require any solvent [40,41]. In one study, Lee et al. examined the effect of GNP nanofiller incorporated into a polyurethane (PU) polymer matrix, where 0.1 wt% graphene loading resulted in 67.2 MPa of tensile strength and 10.60 MPa of Young’s modulus [42]. In another study, Patel et al. investigated the effect of GNP on the shape memory and mechanical properties of microwave radiation-induced TPU, where 2% GNP improved 150% of recovery strength, 50% of constant strain recovery, and 20% of tensile strength [43]. On the other hand, Ivanov et al. studied the effect of GNP on the thermal and electrical conductivities of PLA matrix, where 6 wt% loaded GNP resulted in improvements of 181% in thermal conductivity and a massive increase in electrical conductivity over almost 7–8 decades [44]. Furthermore, Nordin et al. studied the effect of GNP on a PLA/TPU binary polymer matrix, where 1 wt% GNP loading achieved 50 MPa of tensile strength [39]. Consequently, there is a direct effect of graphene being a nucleating agent together with its reinforcing agent on the properties of both single and binary TPU/PLA polymer matrices. Although there are numerous published works on graphene integration in thermoplastic composites in the literature, there is still a lack of knowledge about the aggregation problem based on the selection of graphene type and processing technique and the scalability of the selected compound. This work carries novelty to tailor the structural, thermal, and shape memory properties of a hybrid composite structure with the integration of a biobased polymer in the TPU matrix by reinforcing graphene nanoplates from waste sources and increasing the biobased degree by combining circularity and sustainability approaches at high manufacturing levels. In the current work, the authors select the platelet version of graphene instead of the sheet form to enhance the degree of exfoliation through polymer chains at the melt phase. This work differs from previously published work by applying a scalable approach by using a high-shear thermokinetic mixer providing higher shear rates such as 4000–4500 rpm compared to the conventional extrusion process working at a maximum of 350–600 rpm. Graphene sheets can be dispersed through the polymer chains under high shear rates by a thermokinetic mixer to achieve the uniform distribution of graphene sheets at various nanoparticle loadings of 0.5, 1.0, and 2.0 wt%, as explained in the manuscript.

Therefore, the present study investigated the synergistic effect of graphene in platelet form produced from waste tires with a PLA matrix, and their blend with TPU was manufactured to attain high-performance polymers in terms of mechanical properties and to reach the highest biobased degree with PLA. First, PLA/GNP composites were produced to understand how GNP works with PLA since the PLA phase with GNP is responsible for the mechanical properties in terms of flexural strength, flexural modulus, tensile stress, and tensile modulus, while the TPU phase is responsible for the shape memory property and the elasticity of the polymer. Therefore, it is very important to ensure that the GNP content yields better mechanical results with PLA and then mix the GNP content with the optimized blend ratios. Thus, a single PLA/GNP composite or a binary blend composite PLA/TPU/GNP can be compared in terms of mechanical and dynamical mechanical responses apart from shape memory behavior and elasticity. A thermokinetic mixer was used at high shear loadings to enhance the polymer chain movements and networking with waste tire-driven graphene nanoplatelets (GNP), which enhanced the distribution of GNP through the polymer chains by minimizing the aggregation problem and providing a scalable and sustainable manufacturing process. Optimum GNP loadings were analyzed for neat and blended forms of PLA by performing a comprehensive mechanic, thermodynamic, and thermal analysis. The shape memory property of the sustainable and biobased GNP-integrated PLA/TPU composite was analyzed by calculating shape fixity and shape recovery together with cycle time by defining the thermal characteristics of the developed composite structure.

## 2. Experimental Section

### 2.1. Materials

Polylactic acid (PLA, Luminy L105, Total Corbion, Istanbul, Turkey) is a biobased polymer derived from a natural source of high-heat and high-crystalline PLA with a molecular weight of 105 kDa, a density of 1.24 g/cm^3^, and a melt flow index (MFI) of 70 g/10 min at 210 °C, containing >99% L isomer type in white pellet form. Thermoplastic polyurethane (TPU, Estane 58206, Lubrizol-IMCD, Istanbul, Turkey) is an aromatic polyester-based polymer with high flexibility and high crystallinity with a density of 1.20 g/cm^3^, a hardness of 85 Shore A, and in the form of purple pellets. The FT-IR spectroscopy result is provided in the Appendix A. Graphene nanoplatelets (GNP) are produced by upcycling and recycling processes from waste tires supplied by Nanografen Co., Istanbul, Turkey. GNP with an average platelet size of 50 nm has 9% surface oxygen groups acting as both compatibilizers and reinforcing agents during polymer processing. A detailed characterization of GNP is provided in the Appendix A. 

### 2.2. GNP-Reinforced PLA-Based Composite Manufacturing by Thermo-Kinetic Mixer

Nanocomposites were prepared by using a Dusatec custom-made Gelimat thermokinetic mixer (Dusatec Inc., Fairfield, NJ, USA) and applied at a high shear rate at around ~4500 rpm and 255 °C for 1 min. TPU and PLA pellets were dried in an oven before the melting process at 100 °C for 4 h and 5 h, respectively. The loading ratios of 0.5, 1, and 2 wt% GNP were adjusted and integrated with the PLA polymer at the melt phase to acquire good distribution and supply a high degree of exfoliation to the polymer matrix. PLA and TPU blending ratios were optimized as a function of GNP content to attain high mechanical performance. After the blending process, composites were crushed into small particles and then injection molded by using mini-injection molding equipment (Xplore, Sittard, Holland) for mechanical characterization.

### 2.3. Characterization

The characteristics of GNP were analyzed by using the Renishaw Raman microscope (New Mills, Wotton-under-Edge, Gloucestershire, GL12 8JR, UK) and X-ray diffraction (XRD) technique by using the D2 PHASER Desktop diffractometer (Bruker, Billerica, MA, USA) employing CuKα radiation, and JEOL JEM-ARM200CFEG UHR-TEM (JEOL, Gute Änger 30, 85356 Freising, Germany). Thermal characteristic analysis was investigated with differential scanning calorimetry (DSC) 3 + 700 (Mettler Toledo, Columbus, OH, USA) under nitrogen gas from −80 °C to 200 °C for TPU composites and from 0 °C to 200 °C for PLA composites with a 10 °C min^−1^ heating rate. STARe software (Mettler Toledo DSC 3+/466 Module, Mettler Toledo, Columbus, OH, USA) was utilized to quantify melting enthalpy (Δm), crystallization enthalpy (Δc), glass transition temperature (T*g*), melting temperature ™, and crystallization temperature (T*c*). Dynamic–mechanical property analyses were conducted by using the DMA1 dynamic mechanical analyzer (DMA) (Mettler Toledo, Columbus, OH, USA) via a single cantilever bending from −80 °C to 75 °C with 1 Hz frequency and a 3 Kmin-1 heating rate. Storage modulus (E’), loss modulus (E’’), and tanδ values were calculated by using STARe software.

Shape memory property analysis was conducted by using the DMA1 dynamic mechanical analyzer (DMA) (Mettler Toledo, Columbus, OH, USA) from 25 °C to 60 °C. A dog-bone-shaped sample was used for shape memory tests. The test was performed in the following steps: (1) recording the length of the sample at room temperature, (2) 8 N force was applied to the sample at Ttrans (εm), (3) cooling to 25 °C with loading, (4) unloading the force for fixity (εf) and recording the length, and (5) reheating to Ttrans, triggering the sample for recovery (εr). The shape fixity and recovery ratios were numerically calculated using the below equations [45].
(1)$$\mathrm{Shape}\;\mathrm{fixity}\;Rf=\frac{\varepsilon_{f}}{\varepsilon_{m}}\times 100\%$$
(2)$$\mathrm{Shape}\;\mathrm{recovery}\;Rr=\frac{\varepsilon_{m}-\varepsilon_{r}}{\varepsilon_{m}}\times 100\%$$

Mechanical characteristics, including flexural and tensile properties, were examined according to the ISO 178 three-point bending and ISO 527-2 methods by a 5982 Static universal testing machine (UTM, Instron, Norwood, MA, USA) with a 5 kN load cell through 5 repeating cycles. In-plane thermal conductivity measurements were supplied by thermal conductivity ratio analysis (TPS2500 S, Hot Disk AB, Goteborg, Sweden) at room temperature over 6 repeated cycles. Surface morphologies of GNP and cross-sectional analysis of the produced composite specimens were examined with the Leo Supra 35VP field-emission scanning electron microscope (FE-SEM, Carl Zeiss AG, Jena, Germany) using different voltages.

## 3. Results and Discussions

### 3.1. The Effect of GNP Content on the Mechanical Performance of PLA Nanocomposites

The effect of GNP reinforcement on PLA polymer chains and filler dispersion was examined by analysis of mechanical properties to determine the optimum GNP loading concentration. Herein, different GNP loading ratios of 0.5–2 wt% were incorporated into PLA chains by applying high shear stress via a thermokinetic mixer. Figure 1 and Table 1 show the flexural and tensile stress–strain curves for PLA/GNP nanocomposites. It is seen that the highest flexural strength results in 0.5 wt% GNP-loaded PLA with 87 MPa compared to the neat PLA. Such improvements can be due to the synergetic interactions between polymer chains and nanomaterials, creating an interphase. Interfacial interactions lead to lessening the chain mobility so that higher energy is required for transition. Another reason leading to improvements could be filling the empty pores inside the polymer matrix, thus having favorable interactions between polymer chains and nanomaterials. The reduction of tensile strength was due to the laminar structure of the polymer. Dislocations present in the structure act as stress raisers, so dislocation slip occurs, and the failure is initiated at the end [40]. This phenomenon is related to the internal energy change under tensile loadings. The movement of atoms from their equilibrium position requires less stress and energy. Therefore, the decrease in tensile strength might stem from the polymer chain orientations under stress loading and the movement of the graphene platelet structures, leading to an increase in stress concentrations during elongation [46]. According to the performance of the neat PLA, there is a slight decrease in tensile modulus and tensile strength at 0.5 wt% loadings. As the GNP content increased up to 2 wt%, the tensile strength of PLA composites decreased due to the random alignment of platelet structures through the polymer chains and increasing stress concentrations [47]. For the 2 wt% GNP, the errors seem large probably due to the ineffective dispersion of 2 wt% GNP. Overall, in the GNP-reinforced PLA composites, a slight change in flexural strength and modulus was observed. These consequences can be explained by at the lowest loading, the incorporation of GNP into the PLA matrix was inefficient to show effective stress transfer. On the other hand, at the highest loading, the nanofiller concentration was too massive, leading to agglomeration and a smaller improvement in flexural modulus. Therefore, 0.5 wt% GNP is feasible for use in the PLA/TPU blend system, which will be discussed in the next section.

### 3.2. The Effect of GNP on the Mechanical Performance of PLA/TPU Blend Composite Systems

The effect of GNP nanofiller was investigated in single PLA polymer matrices, and 0.5 wt% was chosen as a loading ratio for use in the blend systems. In order to attain the optimum ratio of PLA and TPU in the blending process, PLA/TPU = 9:1 and PLA/TPU = 8:2 composites were homogeneously obtained after the mixing process in the thermokinetic mixer. A TPU amount higher than 30% did not provide a homogeneous property and was not mixed with PLA, and thus the TPU ratio was kept as low as possible. It is difficult to mix it under high shear rates because the TPU wraps onto the screw of the mixer. Consequently, it becomes compelling to remove the polymer from the mixer. Therefore, this kind of blend ratio can vary according to the different production parameters. Additionally, the increasing content of TPU leads to a reduction in strength and modulus due to the soft nature of TPU. Figure 2 shows the flexural and tensile stress–strain curves and the strength-to-modulus changes of blend composites as a function of the TPU ratio. It is clear that TPU content reaching 20 wt% from 10 wt% leads to a sharp reduction in modulus and strength. The presence of TPU in the blend composite is important to adjust the elasticity and gain shape memory properties. As expected, the unfilled PLA has a much higher flexural and tensile modulus compared to the neat TPU since the stiffness of PLA is dominant. TPU has elevated strain values due to its elastomeric structure, and it may contribute to PLA’s brittleness, as presented by the minor phase in the PLA/TPU blend. Additionally, PLA’s biodegradability can be combined with TPU’s elasticity by using a PLA-to-TPU ratio of 9:1 by weight in the composite structure.

After the selection of PLA/TPU = 9:1 composition, GNP nanoparticles were added in the loading ratios of 0.5, 1, and 2 wt%. Figure 3 and Table 2 represent the flexural and tensile stress–strain curves and mechanical results of PLA/TPU = 9:1 composites at different GNP loadings. The improvement percentages of GNP-reinforced PLA/TPU = 9:1 composites are also presented. The tremendous improvement in flexural strength and flexural modulus belonged to the 0.5% GNP loading with 24% and 10% improvements, respectively, compared to the blend of PLA/TPU = 9:1, which confirms the most effective stress transfer has occurred at that ratio. On the other hand, the 2 wt% GNP loading resulted in less improvement due to the possible agglomeration of high amounts of GNP. Enhancing flexural properties offers insights into the homogeneous distribution of GNP filler, effective stress transfer from the polymer matrix to GNP, and microphase separation acting as a nucleating agent [42]. The nucleating agent leads to early crystallization and increasing total crystallinity [47]. In addition, as the Rule of Mixture suggests, interface occurrence is a crucial point in improving the mechanical properties because as the interface occurs, the volume fraction of the overall composite changes, so the total load is divided relative to the volume fractions of each matrix, nanomaterial, and interface. Thus, the total strength and the total modulus of the composite increase [48]. Overall, the addition of TPU and GNP made the blend more elastic and less brittle; therefore, GNP-loaded composites can overcome higher stresses and respond to higher strains. 

Tensile tests were also carried out to gain full insight into the obtained composites. The tensile properties of blended GNP-incorporated nanocomposites are shown in detail in the Appendix A. In terms of tensile properties, the highest improvement in tensile strength was recorded by the 0.5 wt% GNP-loaded PLA/TPU = 9:1 blend among the nanocomposites at 51.55 MPa. There was a slight decrease in the tensile modulus of PLA/TPU = 9:1+ GNP-loaded composites compared to the PLA/TPU = 9:1 blend structure. These slight changes were due to the dislocations of the composite structure and configuration of polymer chains and nanofillers, leading to a reduction in the tensile strength and modulus of the nanocomposites. Due to dislocations, slip occurred at these stress points, and failure initiated and resulted in fracture. Highly regular ordered laminar structures cause high density and, therefore, high brittleness. The elongation at break percentage was enhanced by the addition of GNP filler since GNP makes the polymer matrix less stiff, so the polymer matrix can overcome applied force and response at higher elongations.

In summary, one of the aims of this study is to investigate the effect of waste-tire-derived GNP on the mechanical behavior of the studied polymers. GNP is a material with a very high surface area, and it is a material with a platelet structure that contains a small number of functional groups. It may not be possible to see similar changes for each property (tensile modulus, tensile strength, flexural modulus, or flexural strength) with these added GNP ratios since they belong to the nanomaterials group, and their acting mechanism is completely different compared to the micrometric or millimetric scale. For example, PLA is a very brittle material and already has a high elastic modulus. Adding 0.5 wt% of GNP to its content may cause an increase in strength values, even if it reduces the modulus values. However, when the loading exceeds 0.5 wt% and reaches 1 wt%, it may cause a decrease in strength values while stiffness starts to have a greater effect. A linear balance is not always achieved with increasing reinforcer ratios, and it may also be a phenomenon related to the intrinsic properties of GNP. The authors think that this should also be investigated. Therefore, the rate of incorporation of such a material into the composite is of great importance, and it might be an option to increase the GNP content more slowly, such as 0.1 wt% to 0.2 wt% and to 0.3 wt%, rather than increasing the content by 2 times.

In this study, to form a blend composite system, the authors declared in the Introduction (Section 1) that PLA/GNP composites are produced primarily to understand how GNP works with PLA, and this choice seems coherent since Figure 3 and Table 2 show that 0.5 wt% GNP was the highest in terms of flexural properties in the PLA/TPU blend matrix.

### 3.3. Thermal Properties, Dynamic Mechanical Behaviors, and Shape Memory Characteristics of GNP-Reinforced PLA/TPU Blend Composites

The thermal properties of composites are important from many perspectives, including determining the compatibility of polymer chains and nanoparticles and the response of composites to thermal energy. The thermal properties of GNP-integrated blend composites are given in Figure 4 and Table 3 below. The compatibility of polymer blends can be predicted via the location of the melting temperatures. The melting temperature of the compatible polymer blend should be in between each polymer’s melting temperature, or the Tm of the minor phase should be closer to the major phase. In these blends and blend composites (Figure 4), PLA is a major phase, and it is dominant since 90 wt% of the blend is PLA while 10 wt% is TPU, which simultaneously makes TPU a minor phase. According to Appendix A, PLA shows a double melting peak with a tiny endothermic peak next to the main melting peak of PLA, which appears at 177.7 °C. On the other hand, the Tm of neat TPU appears at 163 °C, as shown in Appendix A and Appendix A. In Figure 4, two melting peaks are seen and named Tm_1_ and Tm_2_. Explicitly, Tm_2_ belongs to PLA in both curves. However, it is possible that Tm_1_ belongs to either PLA or TPU or that there is an intersection of the two melting peaks. Tm_2_ of the PLA/TPU blend is 175.4 °C, whereas Tm_2_ of PLA/TPU/GNP is 177.4 °C. In the blend composite system, Tm_1_ and Tm_2_ shifted 7 degrees and 2 degrees right, respectively. As seen, Tm_1_ and Tm_2_ of the blend composite structure are nearly in between the melting temperatures of neat PLA and neat TPU, and Tm_1_ becomes closer to Tm_2_ in the blend composite. This balanced melting temperature proves that the polymer blend has compatible structures. By 0.5 wt% GNP integration into the PLA/TPU = 9:1 blend polymer, Tg was improved from 64 °C to 69 °C, Tm was improved from 175 °C to 177 °C, and Tc was improved from 106 °C to 110 °C, as presented in Table 3. These improvements in temperatures prove that GNP fillers lead to crystallizing amorphous segments; thus, crystallization enthalpy increases [49]. Considering the obtained temperature values of the PLA/TPU = 9:1 +0.5% GNP composite, the addition of GNP enhanced the thermal properties of the composites since GNP acts as a nucleating agent and improves the crystallization during injection molding. In other words, the nucleating agent is placed in between polymer chains, thus creating an interface and starting crystallization earlier. This leaves more time to crystallize, resulting in increased crystallization [50]. The increase in enthalpies indicated that GNP nanofillers restrict the movement of polymer chains by integrating GNP so that the energy required to change the phase of the composite is increased [51]. The obtained melting temperature results prove the compatibility of the PLA/TPU = 9:1 blend since the newly formed melting temperatures of the blend system appeared between the individual melting temperatures of PLA and TPU. Further characterization regarding the DSC of TPU- and PLA-based composites as a function of GNP content are presented in the Appendix A. A standard sample with 100% crystallinity was used to calculate the enthalpy [52].

The viscoelastic property of polymers is studied by dynamic mechanical analysis, in which a sinusoidal force is applied to a material, and the resulting displacement is measured as a function of time and temperature. DMA is a significant characterization method in shape memory polymers since dynamic mechanical properties are the underlying feature when a material is expected to show reversible deformation. The mechanical performance of a shape memory polymer under an oscillating load should be monitored to make durable materials. In this study, TPU was used as a shape memory source in a PLA/TPU blend, and therefore, the elastic and viscous behavior/response of the composites were examined. Figure 5 presents the storage modulus, loss modulus, and tan delta values, including the Tg of materials, as a function of temperature. In Figure 5a, the temperature-dependent storage modulus of PLA/TPU = 9:1 is higher than the neat PLA, meaning that TPU increased the energy stored in the material in the same temperature interval. By the incorporation of 0.5 wt% GNP, the storage modulus increased further, and the blend composite system became stiffer by raising its storage capacity. The increase in storage modulus implied that the force between the polymer matrix and GNP particles was empowered due to the enhancement of interfacial adhesions. In Figure 5b, the loss modulus as a function of temperature is seen. Loss modulus and tanδ improvements of the blend composite system indicated an increase in the potential of dissipated energy and also damping capacity, confirming more ultimate internal friction [53]. It was observed that the composite structure increased energy dissipation, leading to an increase in the vibration mode, making the dislocation movement of polymer chains easier. A higher loss modulus is due to a rise in internal friction, leading to an improvement in the dissipation of energy. Above the Tg, the damping factor increases in the transition region since polymer chains are able to move [54]. Below the Tg of composites, loss modulus and energy dissipation tend to decrease since the polymer chains are even more frozen [54]; therefore, overall, it can be said that GNP particles made the composite structure more rigid and left less space for transfer energy. Additionally, due to the internal energy change verified by DMA analysis, the dislocations move faster, so less stress is required to tear the composite structure [46]. Figure 5c,d shows the damping factor and the glass transition temperatures of the materials. The position of the tan delta peak where the storage modulus decreases dramatically and the loss modulus reaches a maximum is used to characterize the glass transition temperature of the materials. The Tg shifted to lower temperatures for the blend since TPU is a highly soft material compared to PLA. On the other hand, the addition of GNP resulted in a shift in higher temperatures due to the enhanced compatibility of the PLA/TPU blend and the rigidity of GNP. The differences in Tg values between DMA and DSC might originate from the different parameters of testing, such as heating rate, removing thermal history, and environmental gas, since all of these have an effect on the crystal alignment of polymers.

Thermal conductivity is an important parameter in the determination of the thermal properties of a material and the investigation of a material’s intrinsic properties regarding how fast heat can be transferred as energy by the oscillation of atoms, as well as the crystallization status of the polymer. Thermal conductivity can be described as the rate of conducting thermal energy through a unit cross-section area. Herein, polymer blend systems incorporated by nanofillers have been offered as solutions to improve thermal conductivity values [55,56,57]. Figure 6 displays the thermal conductivity values of neat PLA, neat TPU, PLA/TPU = 9:1, and PLA/TPU = 9:1 +0.5% GNP. The thermal conductivity of neat PLA is lower due to a lack of entanglement, voids, and chain ends since they function as phonon-scattering points, resulting in inefficient heat transfer, unlike neat TPU. The thermal conductivity of PLA/TPU = 9:1 was the lowest, while PLA/TPU = 9:1 + 0.5% GNP enhanced the conductivity compared to neat PLA and the blend. In the PLA/TPU blend, phonons prefer to pass the heat through the particle phase, elongating the length of the polymer phase, and thus this situation leads to a reduction in the thermal conductivity ratio due to the two-phase polymer chains in the composite structure [56]. On the other hand, by the incorporation of GNP, the thermal conductivity of the composites was enhanced due to the free motion of phonons of GNP nanoparticles, and this is an important parameter in shape memory polymers to preserve heat energy under different thermal stresses [58].

The investigation of shape memory polymer properties examines the most critical parameters: shape fixity (Rf) and recovery ratios (Rr). Not only the fixity and recovery ratios, but also the speed of the cycle responses in these ratios plays a critical role regarding the activation and continuity of shape change, so they should be considered as well. To compare the effect of GNP on shape memory properties, the 0.5 wt% GNP-integrated composite was chosen due to its mechanical and thermal performances, and the measurements were obtained by a thermomechanical cycling procedure. Figure 7a,b describes the evolution of the Rf and Rr values and the fixing and recovery speed of the PLA/TPU = 9:1 and 0.5% GNP-integrated composite. In addition, the zoomed version of the marked part is shown with black arrows. The Rf value of PLA/TPU = 9:1 was close to 98.5% in the first minute and showed no remarkable change in time. On the other hand, the PLA/TPU = 9:1 + 0.5% GNP-loaded composite structure showed a 99.8% fixity ratio in the first minute and reached 100% fixity in 4 min. It can be concluded that GNP reinforcement led to a dramatic increase in the response speed of the fixity cycle. In the case of the recovery parameter, the shape recovery for the PLA/TPU = 9:1 blend responded to a maximum of 99% recovery ratio. At the end of the cycle, the Rr value of the PLA/TPU = 9:1 blend had a tiny change where the response of the GNP-loaded composite was more effective with a remarkable improvement. GNP attainment to the blend polymer matrix showed the dramatic enhancement of the response of shape recovery in the first minute. At the end of the cycle for the PLA/TPU = 9:1 + 0.5% GNP composite, it possessed 99.5% of the Rr value. The shape memory properties were enhanced since elasticity and energy damping increased due to more dominant internal friction with GNP loading, as proved in the thermomechanical analysis. Moreover, 0.5 wt% of GNP attainment to the PLA/TPU = 9:1 composite was enough to leave space for transformations of segmental structures. Moreover, from the thermal conductivity measurements, it is seen that the upcycled GNP led to extensive improvement in the transfer of thermal energy due to their intrinsic properties. Therefore, the integration of GNP provides fast internal heating in composites compared to unfilled polymer blend systems, and the cycle times of shape fixity and recovery processes can be enhanced by increasing the interfacial interaction between polymer chains and GNP particles.

### 3.4. Cross-Sectional Analysis of GNP-Reinforced PLA/TPU Blended Composites

SEM characterization was performed to understand the blending behavior of TPU and PLA polymers and the interaction of GNP particles with the blend composite given in Figure 8. Figure 8a shows the platelet nature of GNP particles explicitly. The SEM image of neat PLA in Figure 8b indicates that PLA has a smooth surface in the fracture region. However, neat TPU has elongated layers, confirming its ductile behavior shown in Figure 8c. It is clear that PLA and TPU polymers respond to stress at different strain approaches. Figure 8d shows the SEM image of the PLA/TPU = 9:1 blend composite having two polymer phases coming from the smoother part of PLA together with the rough surface of TPU. The co-continuous phases of both PLA and TPU matrices are observed, confirming the formation of immiscible blends [39]. With the addition of GNP to the blend composites, the morphology changed, and the separated phases in the neat polymers merged with each other due to the presence of GNP and their interactions with both polymer regions, as seen in Figure 8e,f. Consequently, the fracture behaviors of neat polymers and the immiscible phases of PLA and TPU polymers were directly changed by adding GNP, resulting in the enhancement of the compatibility of polymers supported by the improvement in mechanical properties of this newly designed GNP-reinforced polymer blend composite.

## 4. Conclusions

The current study succeeded in integrating graphene nanoplatelets obtained from waste tires into a PLA/TPU blend system by using a high-shear thermokinetic mixer to provide sustainable and improved shape fixity and shape memory properties. as well as enhancing the mechanical and thermal behavior of the blend polymeric system. By forming a blend system of PLA/TPU (9:1), the brittle behavior of PLA and the low strength of TPU were overcome by increasing the biobased content, and their nanoblends were produced by integrating upcycled GNP. In this blend, 0.5 wt% GNP showed 24% and 16% enhancement in flexural strength and thermal conductivity, respectively, compared to the neat blend. Additionally, 99.8% of shape fixity and 99.58% of shape recovery ratios were achieved with the new design of the blend composite having an adjusted PLA and TPU ratio reinforced by 0.5 wt% GNP. The polymer blend system with the incorporation of 0.5 wt% GNP resulted in a change in the fracture surfaces of neat PLA and TPU since the compatibility of those polymers was improved. DMA studies showed that blend composites have improved storage and loss modulus. This extensive study offers a new perspective on scale-up biobased polymers by integrating upcycled graphene obtained from waste tires by adopting a circular economy approach and providing multifunctionality on shape memory polymers, which can be used in printing and injection processes in the plastic industry.

## Figures and Tables

**Figure 1 polymers-15-01085-f001:**
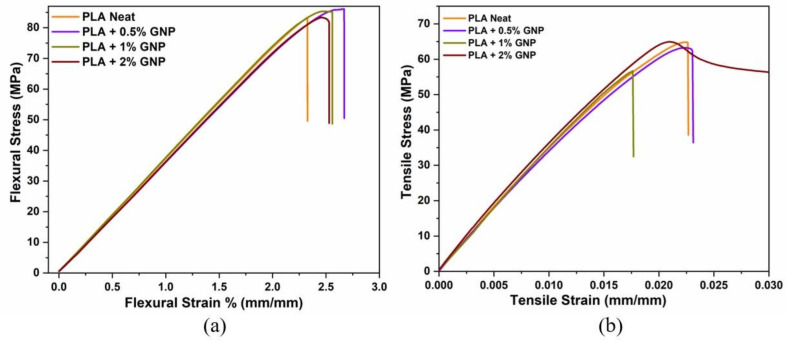
(**a**) Flexural and (**b**) tensile stress–strain curves of PLA/GNP composites.

**Figure 2 polymers-15-01085-f002:**
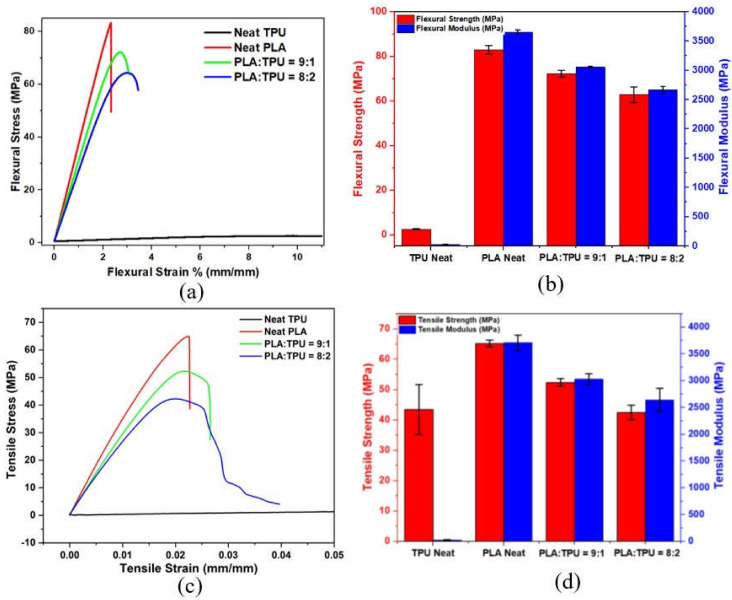
Flexural and tensile (**a**,**c**) stress-strain curves and (**b**,**d**) strength-to-modulus change of PLA/TPU composites based on neat polymers.

**Figure 3 polymers-15-01085-f003:**
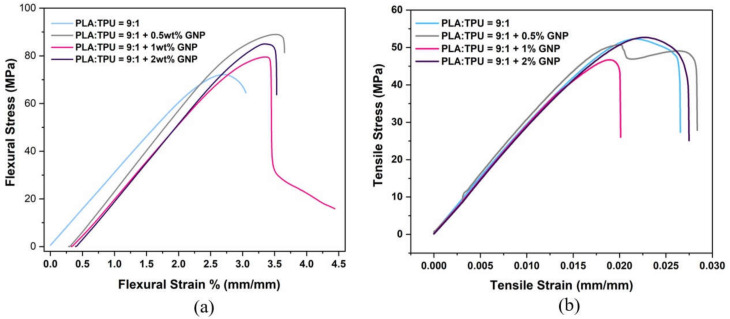
(**a**) Flexural and (**b**) tensile stress–strain curves of PLA/TPU = 9:1 composites at different GNP loadings.

**Figure 4 polymers-15-01085-f004:**
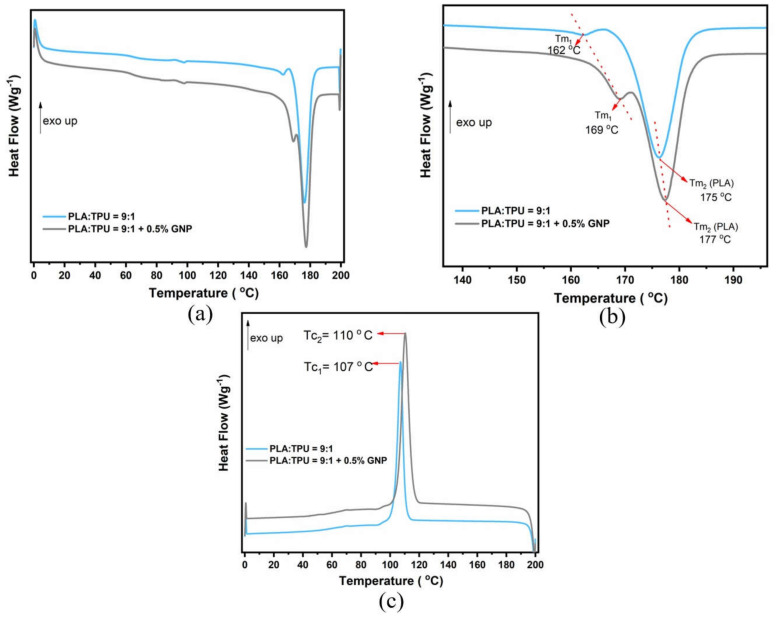
(**a**) Second heating curves, (**b**) zoomed second heating curves, and (**c**) cooling curves of PLA:TPU = 9:1 and PLA:TPU:9:1 + 0.5%GNP.

**Figure 5 polymers-15-01085-f005:**
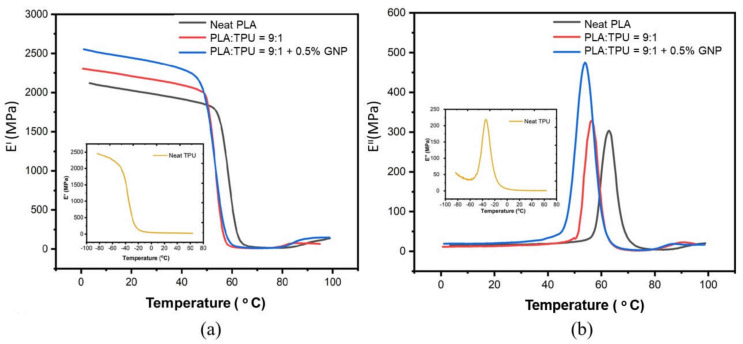
(**a**) Storage modulus, (**b**) loss modulus, (**c**) Tan delta, and (**d**) closer Tan delta values as a function of temperature.

**Figure 6 polymers-15-01085-f006:**
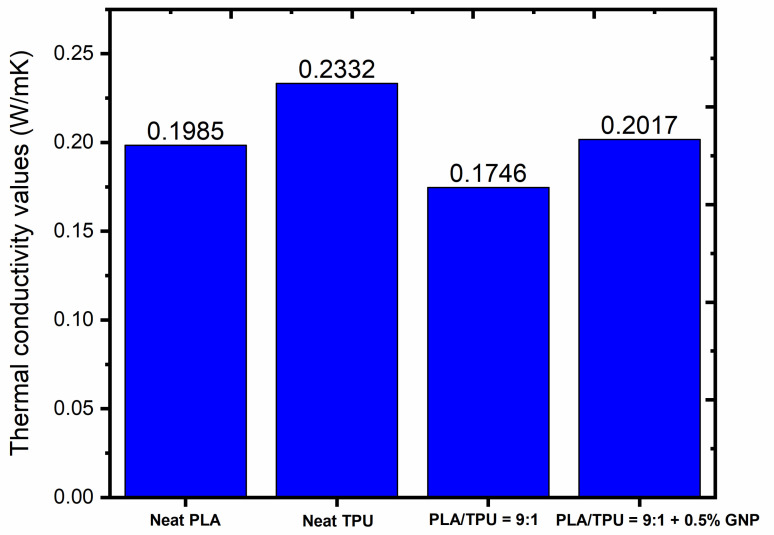
Thermal conductivity values of neat PLA, neat TPU, PLA/TPU = 9:1, and PLA/TPU = 9:1 + 0.5% GNP specimens.

**Figure 7 polymers-15-01085-f007:**
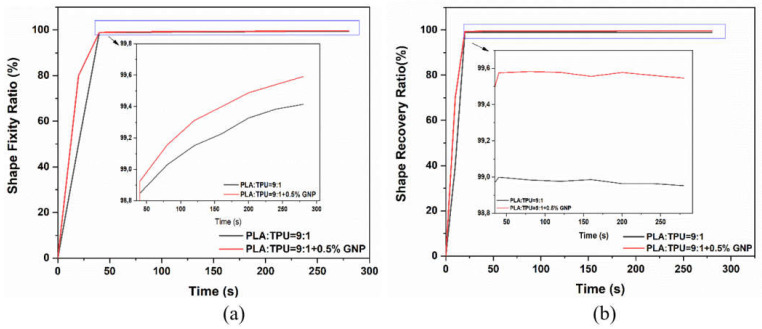
Evolution of (**a**) Rf during the fixing process, and (**b**) Rr during the recovery process of PLA/TPU = 9:1 and PLA/TPU = 9:1 + 0.5% GNP composite.

**Figure 8 polymers-15-01085-f008:**
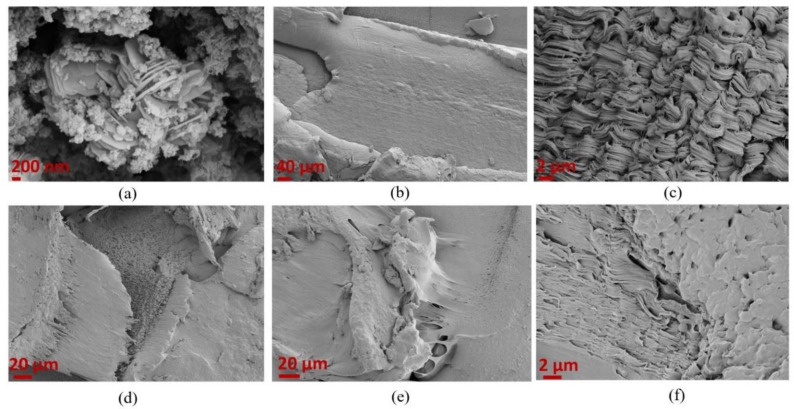
SEM images of (**a**) GNP, (**b**) neat PLA, (**c**) neat TPU, (**d**) PLA/TPU = 9:1 blend composite, and (**e**), and (**f**) PLA/TPU = 9:1 + 0.5 wt% GNP at different magnifications.

**Table 1 polymers-15-01085-t001:** Mechanical properties of PLA-based GNP-loaded composites.

Sample	Flexural Strength (MPa)	Flexural Modulus (MPa)	Tensile Strength (MPa)	Tensile Modulus (MPa)	Elongation at Break (%)
Neat PLA	82.85 ± 2.0	3655.0 ± 33.2	65.20 ± 1.1	3709.5 ± 142.1	3.00 ± 0.6
PLA + 0.5%GNP	86.83 ± 1.5	3527.5 ± 60.2	63.30 ± 0.5	3512.5 ± 317.8	2.85 ± 0.6
PLA + 1%GNP	83.98 ± 1.7	3692.5 ± 27.5	58.78 ± 5.9	3736.8 ± 288.6	2.25 ± 0.7
PLA + 2%GNP	83.45 ± 3.5	3432.5 ± 98.8	60.23 ± 5.5	3542.0 ± 924.9	2.53 ± 1.3

**Table 2 polymers-15-01085-t002:** Flexural properties of neat polymers, blends, and blend composite systems with improvement percentages compared to the reference specimen of PLA/TPU = 9:1.

Sample	Flexural Strength (MPa)	Flexural Modulus (MPa)	Tensile Strength (MPa)	Tensile Modulus (MPa)
Neat PLA	82.85 ± 2.0	3655 ± 33.2	65.20 ± 1.1	3709 ± 142.1
Neat TPU	2.53 ± 0.2	28.55 ± 3.6	43.47 ± 8.2	27.46 ± 5.7
PLA/TPU = 9:1	72.20 ± 1.4	3055 ± 12.9	52.35 + 1.2	3031 ± 99.6
PLA/TPU = 9:1 + 0.5%GNP	89.38 ± 3.2	3375 ± 100.3	51.55 + 3.0	2793 ± 62.7
PLA/TPU = 9:1 + 1%GNP	82.38 ± 6.9	3285 ± 73.7	49.60 + 4.0	2858 ± 115.3
PLA/TPU = 9:1 +2% GNP	86.51 ± 2.1	3233 ± 93.0	48.98 + 4.1	2775 ± 183.9

**Table 3 polymers-15-01085-t003:** Thermal properties of PLA:TPU = 9:1 based GNP integrated nanocomposites.

Sample	T*_g_* (°C)	T*_m_* (°C)	ΔHm (J/g)	T*_c_* (°C)	ΔHc (J/g)
PLA/TPU = 9:1	64.1	175.4	−39.1	106.9	32.3
PLA/TPU = 9:1 +0.5% GNP	69.4	177.4	−42.7	110.4	35.1

## Data Availability

The data presented in this study are available on request from the corresponding author.

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
