# Peer review of "Sustainable Engineered Design and Scalable Manufacturing of Upcycled Graphene Reinforced Polylactic Acid/Polyurethane Blend Composites Having Shape Memory Behavior"

_polymers, 2023, doi:10.3390/polym15051085_

Round 1

Reviewer 1 Report

The authors study graphene reinforced polylactic acid/polyurethane blend composites.

Specific comments

As the authors mention in the introduction, there are several works in the literature on the PLA-TPU blend and their composites with graphene. It is not clear if opposite results are reported in these works from the literature or specific topics that might be studied extensively. For this reason, the necessity of this work must be reported carefully.

Since the authors study the effect of GNPs content on the mechanical performance of PLA nanocomposites, they must include in the introduction all the information from the literature on this topic and compare their results with the corresponding ones from the literature.

In the tables presenting mechanical properties’ data, the decimals should not be used since the error bar refer to an integer. Also, the error bar for tensile modulus for PLA+2%GNPs is 25% and for elongation at break is 50%. Why is the error bar so large? The question is, can these measurements be accepted?

According to table 1 for all the different types of measurements, there is no clear influence of GNPs, monotonous increase or decrease. The authors do not comment on the value of tensile modulus for PLA+1%GNPs which is larger than the one of neat PLA. They choose the PLA+0.5% GNPs as feasible to use in the PLA/TPU blend system, using only the value of flexural strength. Why?

The authors mention (3.2) that TPU content higher than 30% did not result in homogenous properties and it was not mixed with PLA, even though they mention otherwise in the introduction, i.e., in the Lai [30] study, regarding the blend PLA/TPU 50/50. How can this be explained?

Since this study is focused on the influence of GNPs in the PLA/TPU blend, the selected content of 0.5 wt.% GNPs should not be decided from the PLA/GNPs system but from the influence of the GNPs concentration on the  PLA/TPU blend matrix.

The presenting mechanical properties’ experimental results must be compared with the corresponding ones from the literature and the ones of PLA/TPU=9:1 and PLA/TPU=8:2 and must be presented in a table. The chosen PLA/TPU=9:1 must be supported in a better way, by comparing all the mechanical properties’ results, especially the ones that are not presented.

Tables 2 and s4 must be connected and included in the text. The experimental values of elongation at break for PLA/TPU-GNPs present an error bar in the range of 40-50%. Why?

According to Figure S4a, PLA presents a double melting peak. In Figure 4a, the melting plots of PLA and TPU must be added for comparison reasons. Since PLA presents a double melting peak, why is the first peak referred to as the TPU melting peak in figure 4b? According to table S2, the main melting temperature of PLA is 177.65oC and of PLA/TPU=9:1+0.5%GNPs is 177.36oC, which are practically the same.

One decimal must be used for all the measurements presented in tables referring to the thermal properties.

Page 9. Figure 4 must be replaced with Figure 5 in the last paragraph and on page 10 also.

Comparing the values of DSC and Figure 5d, it seems that the Tg value is completely different.

Generally, the authors do not compare their results with the corresponding ones from the literature. 

Author Response

Dear Editors,                                                                                               26 January 2023

We would like to express our sincerest gratitude to you and the referees for the positive and constructive comments, which we wholeheartedly believe that these comments have improved the scientific level and readability of the manuscript significantly. The statements typed in red are those addressing the place of the modifications in the revised manuscript or including additional clarifications on the comments of the respected reviewers in the rebuttal letter.

RESPOND TO REVIEWER 1

Comments and Suggestions for Authors

The authors study graphene reinforced polylactic acid/polyurethane blend composites.

Specific comments

As the authors mention in the introduction, there are several works in the literature on the PLA-TPU blend and their composites with graphene. It is not clear if opposite results are reported in these works from the literature or specific topics that might be studied extensively. For this reason, the necessity of this work must be reported carefully.

Dear reviewer, we include and empower the necessity of this work by adding extra information at the end of the introduction as:

’’ …Although there are numerous published works about graphene integration in thermo-plastic composites in literature, there is still lack of knowledge about the aggregation problem based on the selection of graphene type and processing technique and the scalability of the selected compound. This work carries novelty to tailor the structural and thermal and shape memory properties of hybrid composite structure with the integration of bio-based polymer in the TPU matrix by reinforcing graphene nanoplates coming from waste sources and increasing bio-based degree by combining circularity and sustainabil-ity approaches at high manufacturing levels.  In the current work, authors especially se-lect the platelet version of graphene instead of sheet form in order to enhance the degree of exfoliation through polymer chains at melt-phase. This work differs from the published work by applying scalable approach by using high shear thermokinetic mixer providing higher shear rates such as 4000-4500 rpm compared to the conventional extrusion process worked at maximum 350-600 rpm. In this work, aggregation problem during melt-mixing process is resolved by the presence of surface functional groups of GNP coming from the process directly and there are no additional treatments required to attach the functional groups on the surface of graphene to enhance interfacial interactions with the polymer chains. Graphene sheets can be dispersed through the polymer chains under high shear rates by thermokinetic mixer to achieve uniform distribution of graphene sheets at various nanoparticle loadings of 0.5, 1.0 and 2.0 wt% as explained in the manuscript. However, there are still needs in the development of shape memory properties of biodegradable PLA/TPU blends with tailorable mechanical properties and their interphase enhancement by high degree of graphene exfoliation during melt-mixing…’’

Since the authors study the effect of GNPs content on the mechanical performance of PLA nanocomposites, they must include in the introduction all the information from the literature on this topic and compare their results with the corresponding ones from the literature.

Dear reviewer, we report the synergistic effect of integrate graphene nanoplatelets obtained from waste tires into a PLA/TPU blend system. We started with the PLA/GNP composites to follow an experimental schedule. In the introduction part, we presented some of the studies from PLA/GNP, TPU/GNP and PLA/TPU/GNP to give the readers a perspective as below:

‘’…In one of the studies, Lee et al. examined the effect of GNP nanofiller incorporated into a polyurethane (PU) polymer matrix, and 0.1 wt% graphene loading resulted in 67.2 MPa of tensile strength and 10.60 MPa of Young’s modulus [42]. In another study, Patel et al. in-vestigated the GNP effect on the shape memory and mechanical properties of microwave radiation-induced TPU; 2% GNP improved 150% of recovery strength, 50% of constant strain recovery, and 20% of tensile strength [43]. On the other hand, Ivanov et al. studied the effect of GNP on the thermal and electrical conductivities of PLA matrix; 6 wt% loaded GNP resulted in improvements of 181% in thermal conductivity and a massive increase in electrical conductivity [44]. Furthermore, Nordin et al. studied the effect of GNP on the PLA/TPU binary polymer matrix, and 1 wt% GNP loading achieved 50 MPa of tensile strength [39]….’’

In the tables presenting mechanical properties’ data, the decimals should not be used since the error bar refer to an integer. Also, the error bar for tensile modulus for PLA+2%GNPs is 25% and for elongation at break is 50%. Why is the error bar so large? The question is, can these measurements be accepted?

Dear reviewer, not all the data were presented as bar chart. Therefore, we had to add errors as a text in the table. For the 2%wt GNP, the errors seem large probably due to the ineffective dispersion of 2wt% GNP. When they are added at high amounts they tend to aggregate due to their high surface area and correspondingly the results may differ from each other exceedingly. That is also one of the reason why we shouldn’t reach high amount of GNP. On the other hand, these measurements can be accepted since all the composites were tested under the same conditions. Mechanical characteristics, including flexural and tensile properties, were examined according to ISO 178 three-point bending and ISO 527-2 methods by a 5982 Static Universal Test Machine (UTM, Instron, Norwood, MA, USA) with a 5 kN load cell through 5 repeating cycles.

According to table 1 for all the different types of measurements, there is no clear influence of GNPs, monotonous increase or decrease. The authors do not comment on the value of tensile modulus for PLA+1%GNPs which is larger than the one of neat PLA. They choose the PLA+0.5% GNPs as feasible to use in the PLA/TPU blend system, using only the value of flexural strength. Why?

Since GNP is a material with a very high surface area, and at the same time, it is a material with a platelet structure that contains a small number of functional groups. Therefore, the rate of incorporation of such a material into the composite is of great importance. We tried the ratios of 0.5, 1 and 2. Here, although the loading content increases by 2 times, increases or decreases may not always occur in a uniform way. One of the aims of our study is to investigate the effect of these waste tire-derived GNPs on these polymers. Likewise, it may not be possible to see similar changes for each property (tensile modulus, tensile strength, flexural modulus, or flexural strength) with these added GNP ratios. For example, PLA is a very brittle material and already has high modulus values. Adding 0.5wt of GNP to its content may cause an increase in strength values, even if it reduces the module values. However, when it is exceeded 0.5wt and 1wt% is used, it may cause a decrease in strength values while stiffness starts to have a greater effect. The fact that a linear balance is not always achieved with increasing reinforcer ratios may also be a phenomenon related to the intrinsic properties of GNP, and we think that this should also be investigated. Since GNPs belong to the nanomaterials group, maybe it would be an option to increase the GNP content increments more slowly, such as 0.1wt to 0.2wt to 0.3 wt etc. But in this study, to form a blend composite system, we choose 0.5% wt by looking at the flexural strength values because shape memory polymers are usually under bending force, so flexural strength is of great importance.

The authors mention (3.2) TPU content higher than 30% did not result in homogenous properties and it was not mixed with PLA, even though they mention otherwise in the introduction, i.e., in the Lai [30] study, regarding the blend PLA/TPU 50/50. How can this be explained?

Dear reviewer, there are numerous type of PLA and TPU that differ in hard and soft segments, and different crystallinity content. The polymer raw materials are different with the study of Lai et al. On the other hand, one of the most important parameter is the process conditions. Lai et al produced those blends in the Brabender internal mixer (815605, Plastograph) under 50 rpm in at 170 °C for 10 min. However we produced our blends in a Dusatec custom-made Gelimat thermokinetic mixer and applied at a high shear rate at around ~4500 rpm and 255 °C for 1 min. In our study TPU content higher than 30% did not result in homogenous properties.  When the compound become softer ( here TPU is a very soft material) it is very hard to mix it under high shear rates due to the polymers wrap onto the screw of mixer. It become very difficult to take out the polymer from the mixer. Therefore, this kind of situations can vary according to the different production parameters.

Since this study is focused on the influence of GNPs in the PLA/TPU blend, the selected content of 0.5 wt.% GNPs should not be decided from the PLA/GNPs system but from the influence of the GNPs concentration on the  PLA/TPU blend matrix.

Dear reviewer, influence of the GNPs concentration on the PLA/TPU blend matrix was also shown in Figure 3 and Table 2. In our blend system PLA and TPU are not a miscible blend and the blend ratio of PLA to TPU is (9:1) by weight. Therefore, it is clear that the major phase is PLA in the PLA/TPU blend system and the PLA is mostly responsible for the mechanical properties since TPU is lack of stiffness and strength. Therefore, first we produced PLA/GNP composites to understand how GNP works with PLA. Because PLA phase with GNP will be responsible for the mechanical properties. On the other hand, our choice seems coherent since in Figure 3 and Table 2 show that 0.5wt GNP was the highest in terms of flexural properties in PLA/TPU blend matrix.

The presenting mechanical properties’ experimental results must be compared with the corresponding ones from the literature and the ones of PLA/TPU=9:1 and PLA/TPU=8:2 and must be presented in a table. The chosen PLA/TPU=9:1 must be supported in a better way, by comparing all the mechanical properties’ results, especially the ones that are not presented.

Dear reviewer, TPU is a very soft material and incorporation into PLA results in reduction in mechanical properties. Here TPU was used to enhance shape memory behaviour and also to maintain elasticity. When increasing the TPU content from 10% to 20%, as can be seen in Figure 2, mechanical results are diminishing. Therefore, increasing the TPU content more was not plausible and also we were not able to produce the composite higher than 30wt% due to the process conditions. Since it become so softer, polymer sticks to the screw. We revised the section 3.2 as ‘’…In order to attain the optimum ratio of PLA and TPU in the blending process, PLA:TPU=9:1 and PLA:TPU=8:2 composites were homogeneously obtained after mixing process in thermo-kinetic mixer. The TPU amount higher than 30% did not provide homogeneous property and not mixed with PLA and thus TPU ratio was kept as much as low. Additively, increasing content of TPU lead to reduction in strength and modulus due to the soft nature of TPU.  Figure 2 shows flexural and tensile stress-strain curves and strength to modulus changes of blend composites as a function of TPU ratio. It is clear that TPU content reaching to 20 wt% from 10 wt% leads a sharp reduction in modulus and strength. The presence of TPU in the blend composite is important to adjust the elasticity and gain shape memory property….’’

Tables 2 and s4 must be connected and included in the text. The experimental values of elongation at break for PLA/TPU-GNPs present an error bar in the range of 40-50%. Why?

Dear reviewer, Table 2 and Table S4 merged and have been included in the manuscript upon your suggestion. However, we still prefer to keep the Table s4 to provide % improvements and elongation at break values to avoid too much data in the manuscript as addition. Elongation at break is how much a material can be stretched, as a percentage of its original dimensions, before it breaks. Error bar for elongation at break values in the range of 40-50% probably due to the immiscible structure of PLA and TPU. Since PLA is very brittle and the TPU is very ductile material and they are immiscible, the failure (break) can be located at the points of PLA segments leading low elongation, or at the TPU segment leading high elongation. Therefore, the errors can be higher. The miscibility of the PLA/TPU blends cannot be changed however their compatibility can be enhanced. Here we tried to tailor their compatibility by GNP addition and with the process conditions.

According to Figure S4a, PLA presents a double melting peak. In Figure 4a, the melting plots of PLA and TPU must be added for comparison reasons. Since PLA presents a double melting peak, why is the first peak referred to as the TPU melting peak in figure 4b? According to table S2, the main melting temperature of PLA is 177.65oC and of PLA/TPU=9:1+0.5%GNPs is 177.36oC, which are practically the same.

Dear reviewer, we included the DSC of TPU/GNP composites below. TPU is a thermoplastic elastomer and its’ Tg is around -41 oC and Tm is 163oC as can be seen. GNP integration into PLA:TPU=9:1 blend polymer, Tg was improved from 64 °C to 69 °C, Tm was improved from 175 °C to 177 °C, and Tc from 106 °C to 110 °C. These improvements in temperatures prove that GNP fillers lead to crystallizing amorphous segments; thus, crystallization enthalpy increases.  Considering the obtained temperature values of PLA:TPU=9:1 +0.5% GNP composite, the addition of GNP enhanced the thermal properties of composites since GNP acts as a nucleating agent and improved the crystallization during injection moulding.

Figure: (a) Secondary heating curves and (b) cooling curves of GNP reinforced TPU composites at different loading ratios

Table: Summary of thermal properties of GNP reinforced TPU composites

Sample

Glass Transition Temperature Tg (°C)

Melting Temperature

Tm (°C)

Melting Enthalpy

ΔHM (J/g)

Crystallization Temperature

Tc (°C)

Crystallization Enthalpy

ΔHc (J/g)

Neat TPU

-41.04

163.03

-5.11

89.23

5.76

TPU+0.5%GNP

-42.58

162.34

-6.11

89.37

4.99

TPU+1% GNP

-42.04

162.89

-6.21

89.13

4.58

TPU+2% GNP

-41.69

163.04

-6.48

89.16

5.26

One decimal must be used for all the measurements presented in tables referring to the thermal properties.

We changed it to one decimal for the measurements presented in Table 3 referring to the thermal properties.

Page 9. Figure 4 must be replaced with Figure 5 in the last paragraph and on page 10 also.

Changes have been made.

Comparing the values of DSC and Figure 5d, it seems that the Tg value is completely different.

Dear reviewer, DSC and DMA is a completely different technique to analyse the glass transition temperatures. In DMA, sinusoidal force is applied to a material unlike DSC and therefore the obtained Tg values may differ from the DSC. According to DSC results presented in the manuscript in Figure 4, we can only see the Tg of PLA around 60 oC since the Tg of TPU is below 0oC and we cannot see it our tested temperature interval.

In DMA we can again see the Tg of PLA segment in blend and neat structure (in Figure 5c and 5d.) around 57.5 and 63 oC in Figure 5d. The Tg value of neat TPU was embed in the picture because we were not able to see negative temperatures for blends due to the equipment problems. Therefore comparing the values of DSC and Figure 5d, the Tg of PLA segment in PLA: TPU(9:1) and PLA:TPU=9:1+0.5 % are  64 oC and 69oC according to DSC while they are 57.5oC and 59oC according to DMA results in Figure 5d.

Sincerely yours, 

Assoc. Prof. Dr. Burcu Saner Okan 
Academic Director
Sabanci University
Integrated Manufacturing Technologies Research and Application Center
Composite Technologies Center of Excellence 

Reviewer 2 Report

The authors report the synergistic effect of integrate graphene nanoplatelets obtained from waste tires into a PLA/TPU blend system. Shape memory properties were analysed by defining flexural strength and thermal conductivity.  Properties of the polymers were studied in detail by dynamic mechanical analysis as a function of time and temperature. 

Some minor instances:

-Thermal properties, dynamic mechanical behaviours, and shape memory characteristics of the blend composites were exhaustively discussed. Graphics are explicative. However, I would like to see photos of polymeric blends specimens to visualize the bulky habitus of the materials.

-Spectroscopic analysis is lacking. In addition to FT-IR spectra NMR and absorbance spectra should be provided.

- The results provide inspiration for environment friendly waste/bio-polymer blends useful in a circular economy approach. Therefore, release tests in aqueous environmental conditions should be performed. The authors can draw inspiration from a recent article: DOI:10.3390/molecules25061368 

All in all, the topic is worthy of scientific interest, the article is well packaged, and the conclusions are supported by the experimental data. 

Author Response

Dear Editors,                                                                                               26 January 2023

We would like to express our sincerest gratitude to you and the referees for the positive and constructive comments, which we wholeheartedly believe that these comments have improved the scientific level and readability of the manuscript significantly. The statements typed in red are those addressing the place of the modifications in the revised manuscript or including additional clarifications on the comments of the respected reviewers in the rebuttal letter.

RESPOND TO REVIEWER 2

Comments and Suggestions for Authors

The authors report the synergistic effect of integrate graphene nanoplatelets obtained from waste tires into a PLA/TPU blend system. Shape memory properties were analysed by defining flexural strength and thermal conductivity.  Properties of the polymers were studied in detail by dynamic mechanical analysis as a function of time and temperature. 

Some minor instances:

-Thermal properties, dynamic mechanical behaviours, and shape memory characteristics of the blend composites were exhaustively discussed. Graphics are explicative. However, I would like to see photos of polymeric blends specimens to visualize the bulky habitus of the materials.

Dear reviewer, thank you for your positive comments.  We included the test specimens below. We have included the tensile and flexural test specimens.

-Spectroscopic analysis is lacking. In addition to FT-IR spectra NMR and absorbance spectra should be provided.

Dear reviewer, Spectroscopic techniques such as NMR; FTIR, UV are very important techniques to understand how light communicate with the material therefore we can distinguish different atomic characteristic, vibrations, confirmation of synthesis for various applications in characterization, optoelectronic devices, dyes, pigments, solutions, polymer synthesis etc. However, in our study, our main aim to create a mechanically stable having shape memory polymer by using a nano blending approach. We were mainly interested with their rheological behaviours to understand the melt dispersion, and their mechanical stability under various forces and temperatures. Here, there is a macro scale compounding by blending of 2 polymers with GNP and their thermal, mechanical, and thermomechanical properties as well as shape memory behaviours are the main subject of the paper. We believe that NMR and UV would be supportive but unfortunately, we are not able to proceed NMR and UV for now. However, we will try to keep in mind these comments and try to include more spectroscopy for our next studies.

- The results provide inspiration for environment friendly waste/bio-polymer blends useful in a circular economy approach. Therefore, release tests in aqueous environmental conditions should be performed. The authors can draw inspiration from a recent article: DOI:10.3390/molecules25061368 

Dear reviewer, we would like to thank for the suggestion as well as the article you mentioned. We read the article carefully and it helped us gain different perspectives. We will certainly try to design such experiments for our future works.

All in all, the topic is worthy of scientific interest, the article is well packaged, and the conclusions are supported by the experimental data. 

Authors would like to thank for your positive support and valuable suggestions.

Yours sincerely,

Assoc. Prof. Dr. Burcu Saner Okan

Academic Director of SU-IMC

Research oriented faculty member

Round 2

Reviewer 1 Report

The authors study graphene reinforced polylactic acid/polyurethane blend composites.

Specific comments regarding the authors reply letter:

·     Authors reply: there is still lack of knowledge about the aggregation problem based on the selection of graphene type and processing technique

Comment: If the authors wish to focus on the better dispersion of the prepared materials using a high shear thermokinetic mixer, they should provide evidence of the GNPs dispersion in the selected matrices. Since No characterization of the GNPs' dispersion has occurred, the assumption of well dispersed filler mentioned in this manuscript cannot be expressed.

·       Authors reply: aggregation problem during melt-mixing process is resolved by the presence of surface functional groups of GNP coming from the process directly

Comment: How do you support this assumption (of surface functional groups of GNP coming from the process directly) since no evidence are presented in this study ? If there is no way to prove this, please remove this sentence.

·       Authors reply: We started with the PLA/GNP composites to follow an experimental schedule.

Comment: Please add to the manuscript the reasons why you followed this schedule. Why did you choose to conduct a preliminary study on PLA/GNPs nanocomposites? A more reasonable approach would be to study several PLA:TPU blends (such as 9.5:0.5, 9:1 as you did, 8.5:1.5, 8:2 as you did) and the incorporate various GNPs content in the selected blend. Please add to the manuscript the reason the blends containing larger concentrations of TPU were not studied. The following comment present specific statements that should be included in the manuscript.    

·       Comments: The following authors’ responses must be added in the manuscript

“For the 2%wt GNP, the errors seem large probably due to the ineffective dispersion of 2wt% GNP”.

Since GNP is a material with a very high surface area, and at the same time, it is a material with a platelet structure that contains a small number of functional groups. Therefore, the rate of incorporation of such a material into the composite is of great importance. We tried the ratios of 0.5, 1 and 2. Here, although the loading content increases by 2 times, increases or decreases may not always occur in a uniform way. One of the aims of our study is to investigate the effect of these waste tire-derived GNPs on these polymers. Likewise, it may not be possible to see similar changes for each property (tensile modulus, tensile strength, flexural modulus, or flexural strength) with these added GNP ratios. For example, PLA is a very brittle material and already has high modulus values. Adding 0.5wt of GNP to its content may cause an increase in strength values, even if it reduces the module values. However, when it is exceeded 0.5wt and 1wt% is used, it may cause a decrease in strength values while stiffness starts to have a greater effect. The fact that a linear balance is not always achieved with increasing reinforcer ratios may also be a phenomenon related to the intrinsic properties of GNP, and we think that this should also be investigated. Since GNPs belong to the nanomaterials group, maybe it would be an option to increase the GNP content increments more slowly, such as 0.1wt to 0.2wt to 0.3 wt etc. But in this study, to form a blend composite system, we choose 0.5% wt by looking at the flexural strength values because shape memory polymers are usually under bending force, so flexural strength is of great importance”.

it is very hard to mix it under high shear rates due to the polymers wrap onto the screw of mixer. It became very difficult to take out the polymer from the mixer. Therefore, this kind of situations can vary according to the different production parameters”.

Therefore, first we produced PLA/GNP composites to understand how GNP works with PLA. Because PLA phase with GNP will be responsible for the mechanical properties. On the other hand, our choice seems coherent since in Figure 3 and Table 2 show that 0.5wt GNP was the highest in terms of flexural properties in PLA/TPU blend matrix”.

Other comments for authors reply:

·       On the other hand, these measurements can be accepted since all the composites were tested under the same conditions”. Of course all the measurements have been done with the same way but the question still remains. Why the error bar is so high for tensile modulus and for elongation at break for PLA+2%GNP and not for the other properties? According to your response, this behavior is due to “the ineffective dispersion of 2wt% GNP”. Please add this comment and elaborate this effect. 

·       we included the DSC of TPU/GNP composites below”. No figure has been added in the response letter. The authors have not answer to the comment. A DSC diagram of neat PLA and neat TPU together for comparison reasons should be added. According to the authors Tm of TPU is 163oC. PLA presents a double melting behavior (according to Fig. S4). How did the authors conclude that the second melting peak at lower temperatures is due to the melting of TPU and not the PLA double melting behavior?

·       Of course DSC and DMA are different techniques, but from both of these techniques we can calculate the value of Tg. Of course we can expect some differences at the values of Tg for the same material but it is peculiar the two techniques to give the opposite results, for TPU(9:1) and PLA:TPU=9:1+0.5 % are 64 oC and 69oC according to DSC while they are 57.5oC and 59oC according to DMA results in Figure 5d. This difference must be explained and the explanation cannot be the difference of the methods. Also the comment for this must be added in the manuscript.

Comments for supplementary materials:

·       The plot of TPU must be added.

·       The accuracy of the temperature and the heat flow is lower than two decimals in table S2.

·       According to table S2 and figure S4 there are serious problems with the calculated values. For the crystallization temperature, according to figure S4, the lower to higher crystallization temperatures are of: PLA+0.5% GNP, PLA+1% GNP, PLA and finally PLA+2% GNP. According to table S2 the sequence of the Tc are: PLA, PLA+0,5% GNP, PLA+1% GNP, PLA and finally PLA+2% GNP. The reported in the table crystallization temperatures do not match to the ones on Figure S4. The same issue occurs for the melting temperatures also. According to figure S4 the lower value refers to PLA+1%GNP (around 175 oC and not 177 oC) and according to table S2 the lower value refers to PLA+0.5%GNP. How can these be explained? Please correct these temperatures and comment the materials behavior in the manuscript.

·       In table S3 the column for the ΔHM100 must be deleted and just reported this value in the manuscript.

·       It is wrong to write values with two decimals when the error bar refers to integers. Please correct this to all the tables including mechanical properties (manuscript and supplementary material).

Comments about the text of the manuscript:

·       Abstract: “high degree exfoliation of GNP”. No proof of highly exfoliated GNPs in the selected matrices has been provided, regarding the prepared materials in this study, therefore this assumption should be removed.

·       Throughout the manuscript double pairs of brackets are used, i.e., [30]-[32]. Please use one pair of brackets.

·       In the introduction, “On the other hand, Ivanov et al. studied the effect of GNP on the thermal and electrical conductivities of PLA matrix; 6 wt% loaded GNP resulted in improvements of 181% in thermal conductivity and a massive increase in electrical conductivity [44].”: Be more specific, what is the % increase in electrical conductivity?

·       In the Materials section, “GNP with the average platelet size of 50 nm has 9% surface oxygen groups acting as both compatibilizer and reinforcing agent in the polymer processing.”: Is the average thickness or number of graphene sheets of the platelets known? If yes, please add this info.

·       In the Characterization section: Please add the transmission electron microscope used for the GNPs characterization (as shown in Fig S2a).

In the Results and Discussion section:

·       Figure 3a: All the curves should begin at 0 point, otherwise the results are not credible regarding the strain %. In this diagram, it seems like no stress was applied in the beginning of the measurement for the composite materials. In addition, the reader cannot compare the curves properly regarding the flexural modulus results.

·      In tensile properties, the highest improvement in tensile strength was recorded by 0.5 wt% GNP loaded PLA:TPU=9:1 blend as 51.55 MPa.”: Pease specify that the highest tensile properties values are attained among the nanocomposites, since neat PLA:TPU blend presents higher values among the prepared blend materials.

·       Compared with the precursor of blends involving TPU and PLA polymers, the tensile and flexural properties of blend with GNP, including composites, possessed superior outcomes.”: The flexural properties are indeed improved, the tensile properties however present a slight decrease. Please correct this sentence.

·       Supporting the crystallization of amorphous segments, the crystallization percentage has been calculated and shown in Table S3, given in the supplementary document. A standard sample with 100% crystallinity was used to calculate the enthalpy [52]. According to the crystallinity measurements, the crystallinity percentage of GNP-loaded composites has risen to 48% where neat PLA had 43% of crystallinity.”: Table S3 presents the results of the PLA/GNPs nanocomposites and not of the blend PLA:TPU/GNPs nanocomposites, therefore a definitive result of increased crystallinity cannot be expressed.

·       The thermal conductivity of Neat PLA is lower due to a lack of entanglement, voids, and chain ends since they function as stress concentration points, resulting in inefficient heat transfer unlike Neat TPU.”: Did you mean phonon scattering points? Please correct this sentence.

·       Higher concentration loading of GNP has been might not provide the appropriate emptiness to alter the shape and increased the cycle times.”:  Since there are no measurements to support such statement, this assumption should not be made. Also, please correct the syntax of this sentence.

·       The 3.5. Cross-sectional analysis of GNP reinforced PLA/TPU blended composites, should be moved prior to the mechanical and thermal characterization.

·     In the Conclusion section: “The current study achieved to integrate graphene nanoplatelets obtained from waste tires into a PLA/TPU blend system by using a high shear thermokinetic mixer to provide a well dispersed, sustainable, and improved shape fixity and shape memory properties as well as enhancing the mechanical and thermal behavior of the blend polymeric system.”: As earlier stated, no characterization of the GNPs' dispersion has occurred therefore, the assumption of a well dispersed filler cannot be made.

Author Response

Dear Reviewer,                                                                                      

We would like to express our sincerest gratitude to you and the referees for the positive and constructive comments, which we wholeheartedly believe that these comments have improved the scientific level and readability of the manuscript significantly. The statements typed in red are those addressing the place of the modifications in the first revised manuscript and the statements typed in green are those addressing the latest modification or including additional clarifications on the comments of the respected reviewer in the rebuttal letter.

The detailed document in pdf version is attached. Please check the supporting document for the detailed information of manuscript. 

Sincerely yours,

Assoc. Prof. Dr. Burcu Saner Okan 
Academic Director
Sabanci University
Integrated Manufacturing Technologies Research and Application Center
Composite Technologies Center of Excellence 

Round 3

Reviewer 1 Report

Reviewer 1 comments

·         Regarding the reviewer’s comment in the previous revision round: ”According to the authors Tm of TPU is 163oC. PLA presents a double melting behavior (according to Fig. S4). How did the authors conclude that the second melting peak at lower temperatures is due to the melting of TPU and not the PLA double melting behavior?”. No definitive response was given regarding the comment above. In figure S5, the melting, as well as crystallization enthalpies of neat TPU and its GNPs nanocomposites, have a very small value, and the aforementioned melting peaks (small melting peak of PLA and melting peak of TPU) almost coincide at the same temperatures. Therefore, the above peaks could belong to either PLA and/or TPU’s melting behavior. This possibility should be taken into account and mentioned in the manuscript. Furthermore, the crystallization peak of TPU during the cooling cannot be observed in the cooling curves of the PLA:TPU blends possibly due to the small TPU content in the materials. This is an indication that suggests that most probably, the melting peak of TPU might not be so apparent.

·         Fig 3a. In this diagram, regarding the initial stages of the measurements of the nanocomposites, there are two possibilities: either an experimental error during the measurement, where there is a delay between the flexural stress application and the recording of the strain, has occurred (most probably, it a common error during mechanical testing), either there is a phenomenon responsible for this result. The reviewer suggests to the authors either comment on a possible phenomenon or simply subtract the appropriate value from the strain axis so that the curves begin at 0. Flexural strength and modulus values are not affected by this correction. 

Author Response

10 February 2023

Dear Reviewer, 

.       Regarding the reviewer’s comment in the previous revision round:” According to the authors Tm of TPU is 163oC. PLA presents a double melting behaviour (according to Fig. S4). How did the authors conclude that the second melting peak at lower temperatures is due to the melting of TPU and not the PLA double melting behaviour?”. No definitive response was given regarding the comment above. In figure S5, the melting, as well as crystallization enthalpies of neat TPU and its GNPs nanocomposites, have a very small value, and for mentioned melting peaks (small melting peak of PLA and melting peak of TPU) almost coincide at the same temperatures. Therefore, the above peaks could belong to either PLA and/or TPU’s melting behaviour. This possibility should be taken into account and mentioned in the manuscript. Furthermore, the crystallization peak of TPU during the cooling cannot be observed in the cooling curves of the PLA: TPU blends possibly due to the small TPU content in the materials. This is an indication that suggests that most probably, the melting peak of TPU might not be so apparent.

Response:

Dear reviewer, we changed the Figure 4b by removing the name of the melting peaks and uploaded as an attachment with the new version of Figure 4b(revised). We also changed the image in the manuscript. Below statements are revised and added to manuscript considering your valuable comments and suggestions.

‘’….Thermal properties of the composites are important in many perspectives including determining the compatibility of polymer chains and nanoparticle, and the response of composites to thermal energy. The thermal properties of GNP-integrated blend composites are given in Figure 4 and Table 3 below. Compatibility of polymer blends could be predicted via the location of the melting temperatures. Melting temperature of the compatible polymer blend should be in between each polymer’s melting temperature or the Tm of minor phase should be closer to major phase. In this blends and blend composites (Figure 4), PLA is the major phase, and it is dominant since the 90 wt. % of the blend is PLA while 10 wt.% is TPU which makes TPU minor phase simultaneously. According to Figure S4, PLA shows double melting peak with the tiny endothermic peak next to the main melting peak of PLA which appears at 177.7 oC. On the other hand, Tm of Neat TPU appears at 163 oC as shown in Figure S5 and Table S4. In Figure 4, two melting peaks are seen and named as Tm1 and Tm2. Explicitly, Tm2 belongs to PLA in both curves. However, it is possible that Tm1might belong to either one of the PLA or TPU, or there may be an intersection of two melting peaks. Tm2 of PLA/TPU blend is 175.4 °C where Tm2 of the PLA/TPU/GNP is at 177.4 oC. In blend composite system, Tm1 and Tmshifted 7 degrees and 2 degrees right, respectively . As seen, Tm1 and Tm2 of blend composite structure is nearly in between the melting temperatures of Neat PLA and Neat TPU and Tm1 get closer to Tm2 in the blend composite. This balanced melting temperature proves that the polymer blend has compatible structures [51–53]. By 0.5 wt.% GNP integration into PLA: TPU=9:1 blend polymer, Tg was improved from 64 °C to 69 °C, Tm was improved from 175 °C to 177 °C, and Tc from 106 °C to 110 °C as presented in Table 3. These improvements in temperatures prove that GNP fillers lead to crystallizing amorphous segments; thus, crystallization enthalpy increases [49]….’’ 

  • Fig 3a. In this diagram, regarding the initial stages of the measurements of the nanocomposites, there are two possibilities: either an experimental error during the measurement, where there is a delay between the flexural stress application and the recording of the strain, has occurred (most probably, it a common error during mechanical testing), either there is a phenomenon responsible for this result. The reviewer suggests to the authors either comment on a possible phenomenon or simply subtract the appropriate value from the strain axis so that the curves begin at 0. Flexural strength and modulus values are not affected by this correction.

Response:

Dear editor, we removed the Figure 3 from the manuscript and changed it with Figure 3 (revised) as below. The changes have been made in manuscript. Figure 3(revised) also has been uploaded as attachment.
